# Analysis of the Drawing Process of Small-Sized Seam Tubes

**Alexander Schrek [1,\*], Alena Brusilová [1], Pavol Švec [1], Zuzana Gábrišová [1] and Ján Moravec [2]**

[1]  Institute of Technologies and Materials, Faculty of Mechanical Engineering, STU Bratislava, Nám. Slobody 17, 812 31 Bratislava, Slovakia; alena.brusilova@stuba.sk (A.B.); pavol.svec@stuba.sk (P.Š.); zuzana.gabrisova@stuba.sk (Z.G.)

[2]  Department of Technological Engineering, Faculty of Mechanical Engineering, University of Žilina, Univerzitná 8215/1, 010 26 Žilina, Slovakia; jan.moravec@fstroj.uniza.sk

\*  Correspondence: alexander.schrek@stuba.sk; Tel.: + 421-905-460-855

**Abstract:** This article is focused on an analysis of factors negatively affecting the tube production process of tubes made from austenitic stainless steel with a very small diameter of ϕ 0.34 mm. The analysis was concentrated on factors that affect the drawing process stability of the seam tubes where the desired final dimensions—a diameter of ϕ 0.34 mm and a wall thickness of 0.057 mm—are limiting factors. Seam tubes made from steel 1.4306 and 1.4301, from producers KT and EW with a longitudinal weld line made by tungsten inert gas (TIG) welding, were used as blanks for constituent drawing operations. It is desirable to provide sufficient inert gas flow and cooling during the formation of a weld joint in a protective atmosphere chamber. A significant temperature gradient prevents the formation of undesirable $Cr_{23}C_6$ carbides in the heat-affected zone (HAZ) which negatively affects the plasticity and formability of the steel and is the cause of technological fractures.

**Keywords:** austenitic steels; seam tubes; strain; TIG welding; drawing process

---

## 1. Introduction

Seam tubes made from austenitic stainless steel are used as the workpiece in the production of medical needles. They are produced by a combination of forming and welding technologies. The process includes the following stages: (1) Continual bending of blanks in a sheet metal roll form, with the dimensions 10 mm × 1 mm; (2) Tungsten inert gas (TIG) welding; (3) Drawing over a floating mandrel in the diameter, and thickness reduction of the wall; (4) Tube drawing without a mandrel in the diameter reduction. The drawing processes are continual, and the drawing length can be up to 300 m, depending on the blank dimensions However, technological fractures can occur during the drawing process, negatively impacting production.

The stable technological process runs continually according to predetermined and minimally changeable parameters, for as long as there is a formation of technological fractures that can cause a process interruption. The factors affecting the formation of technological fractures are:

1.  Geometry of functional tools parts: geometry of the drawing die and a mandrel for wall thickness reduction with a floating mandrel (the first two operations) and geometry of the drawing die during tube drawing without a mandrel (subsequent operations) [1–3];
2.  Stress conditions in the compress and size fixing area of the functional tool parts [4–6];
3.  Effect of tool parameters on the drawing process [1,7];
4.  Similarity of the geometric tubes and thick-walled criterion [8–10];
5.  Effect of a weld and the heat-affected zone on tube eccentricity and the drawing process [6,9];

6. Lubrication effect on the drawing process [11];
7. Effect of stress–strain parameters of the formed material on the initial dimensions of the workpiece [12,13];
8. The magnitude of the drawing forces and factors affected their magnitude [13];
9. Parameters of the selected, most commonly used materials.

To analyze the technological process and predict the parameters affecting the continuity of the production process, a simulation of the whole technological process up to the smallest required tube size was run. Knowledge from the stress–strain analysis of the processed materials was used boundary conditions, with emphasis on the magnitude of the true stresses in the deformation zones, the magnitude of the drawing forces and the resulting wall thickness.

The aim of the study was to verify the results of a simulation of the technological process of small-sized tube production with measured and calculated values and analytically define the causes of technological fracture formation. From these presented factors, the focus was concentrated on the effect of the weld on the plastic properties of the formed material.

## 2. Materials and Methods

### 2.1. Material Characteristics

Seam tubes, such as those considered for the production of injection needles, are made of austenitic steels complying with the EN 1.4306, respectively, EN 1.4301 and EN 10088-2 standards. These standards characterize the general dimensional and basic stress–strain characteristics of the material entering the technological process as well as their chemical compositions [12].

The maximal permissible of a burr size on the edge is 10% of the strip thickness. The surface of the starting material should be high-bright, without surface protrusions, scales and pores and with prescribed roughness. The chemical compositions of the materials used for research (EW DIN 17 441 and material KT X2CrNi19-11) are shown in Table 1.

**Table 1.** Chemical compositions of the tested materials.

| Material | Chemical Composition wt % | | | | | | |
|---|---|---|---|---|---|---|---|
| EW | C | Si | Mn | P | S | Cr | Ni |
|  | 0.027 | 0.56 | 1.07 | 0.019 | 0.001 | 18.02 | 10.05 |
| KT | C | Si | Mn | P | S | Cr | Ni |
|  | 0.012 | 0.52 | 1.56 | 0.022 | 0.05 | 18.25 | 10.08 |

The presented steels correspond to the norms EN 10088-2 and DIN 17 441. They were in a cold-rolled state with a high-bright surface.

### 2.2. Manufacturing Process of the Seam Tube

The workpiece is a metal strip with an optimized thickness and width of 0.1 mm × 10 mm, for a final tube dimension of ϕ 0.34 mm × 0.057 mm. The semiproduct of the tube, which is ϕ 3.2 mm was made by means of the technological process outlined in Figure 1. It is rolled up after continual bending and welding. The dimensional changes are achieved through drawing operations on draw benches Bougrad Toolmatic with the purity degree "E". Reduction of the diameter and wall thickness is achieved with a floating mandrel, to the order of ϕ 2.84 mm × 0.075 mm in the first drawing and ϕ 2.52 mm × 0.063 mm in the second drawing. The following operations were used for diameter reduction, by means of tube drawing without a mandrel.

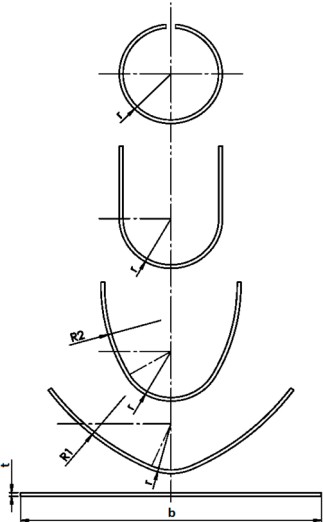

**Figure 1.** Technological process of seam tube production through a bending process.

2.2.1. Welding Assembly

The austenitic stainless steel tubes were arc welded using tungsten inert gas welding without a filler material (TIG 142), as shown in Figure 2. The welding gap dimension was minimized through the calibration pulleys pressure. The aim was to create a completely tight weld without additional material and cracks. Favorable weld root formation was secured using pressure, and also the presence of an inert gas. This way, we created the conditions for a perfect compact weld formation without pores and cavities. Argon was used as the inert gas with a purity of 99.95% and the welding direct current had a straight polarity. In this way, the TIG welded seam tubes were able to be used as workpieces for medical needles production [14,15].

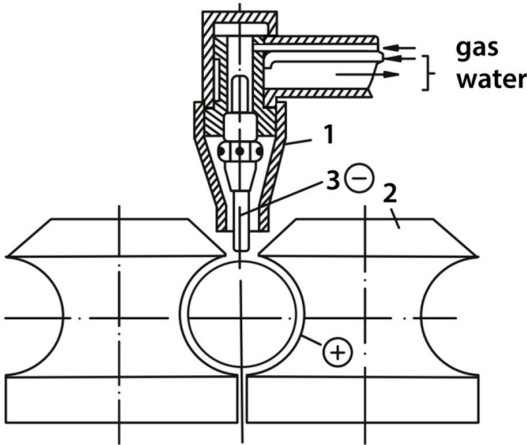

**Figure 2.** Seam tubes welding by the tungsten inert gas (TIG) method: 1—nozzle, 2—calibration pulleys creating additional pressure, 3—non-consumable tungsten electrode.

2.2.2. Stress States during Tube Drawing over a Floating Mandrel and without a Mandrel

The stress states illustrated in the figures characterize:

- Drawing over a floating mandrel (Figure 3), where single strain areas are determined by a reduction part, a calibration part and a part behind the drawing die;
- Drawing without a mandrel (Figure 4), where stresses and strains are determined by a reduction part and a part behind the drawing die.

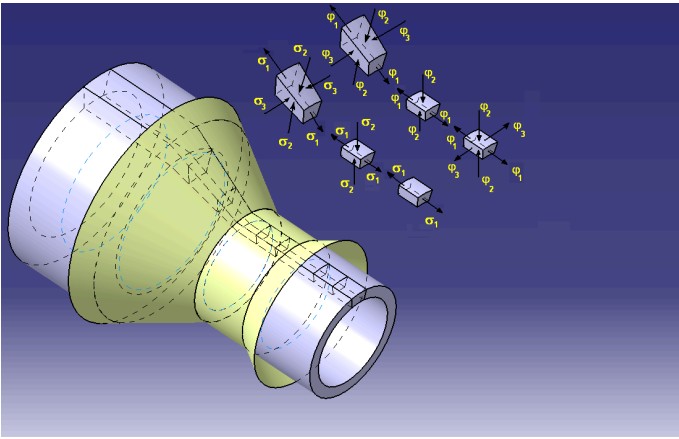

**Figure 3.** The stresses and strains during drawing over a floating mandrel.

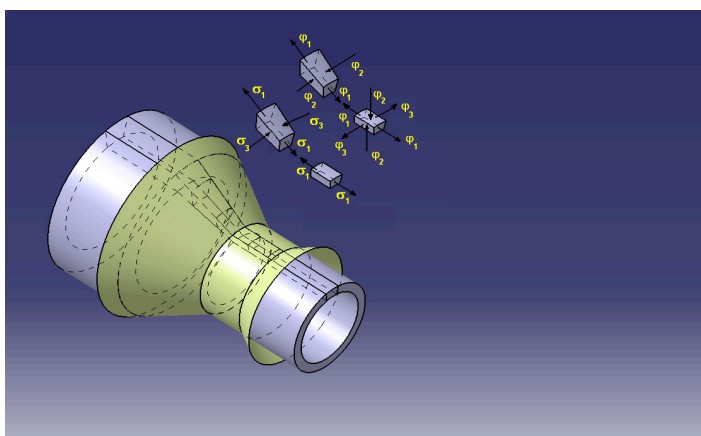

**Figure 4.** The stresses and strains during drawing without a mandrel.

The predominant stress state during drawing over a floating mandrel is a combination of two compressive stresses and one tensile stress that acts in the drawing force direction.

During drawing over a floating mandrel, there is a characteristic equality of main stresses and as well as main strains [10], resulting from volume uniformity at the development of plastic strains. Therefore, it is possible to write:

$$\frac{2 \times \pi \times l \times r \times t}{2 \times \pi \times l_0 \times r_0 \times t_0} = 1 \tag{1}$$

after logarithm

$$\ln \frac{l}{l_0} + \ln \frac{r}{r_0} + \ln \frac{t}{t_0} = 0 \tag{2}$$

$$\varphi_1 + \varphi_2 + \varphi_3 = 0 \tag{3}$$

where

$$\varphi_1 = \ln \frac{l}{l_0} \tag{4}$$

$$\varphi_2 = \ln \frac{r}{r_0} \tag{5}$$

like

$$\varphi_3 = \ln \frac{t}{t_0} \tag{6}$$

where

$$\varphi_2 = -\varphi_3 \tag{7}$$

(initial dimensions of tube: $l_0$—initial length, $r_0$—initial radius, $t_0$—initial wall thickness
final dimensions of tube: l—final length, r—final radius, t—final wall thickness
$\varphi_1$—major strain, $\varphi_2$—minor strain, $\varphi_3$—minor strain)

The tube elongation during drawing over floating mandrel is due to the diameter and wall thickness reduction [16–18]. During drawing without a mandrel, its development of the plane state of stress and single parts are defined by a reduction part and a part behind the drawing die as shown in Figure 4.

## 3. Results

### 3.1. Microstructure Analysis

The microstructure of stainless steel X2CrNi19-11 grade, produced by two companies (KT and EW), was analyzed. The microstructure of the tubes made of stainless steel produced by EW observed in both longitudinal and transversal cross-sections, had a high strained austenitic matrix with a deformation texture and finely dispersed carbide phase as can be seen in the longitudinal cross-sections in Figure 5. The cross-section of the ribbon semiproduct produced by KT is depicted in Figure 6. The microstructure consists of an austenite matrix but without a carbide phase or other microconstituents. The grain size of the matrix was slightly higher compared to the stainless steel produced by EW. However, the austenite grain size of both steels was between 0.015 and 0.025 μm and met the required standards.

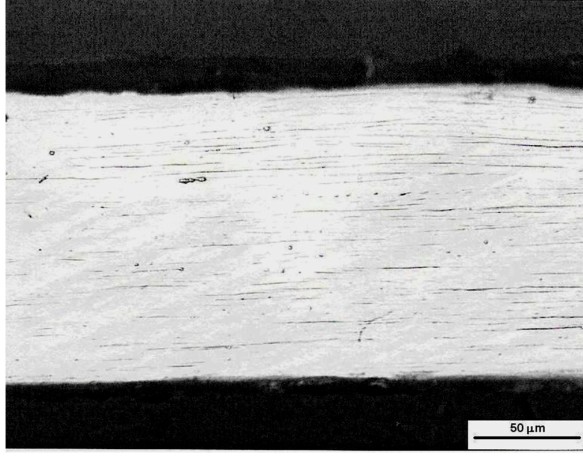

**Figure 5.** The tube EW longitudinal section.

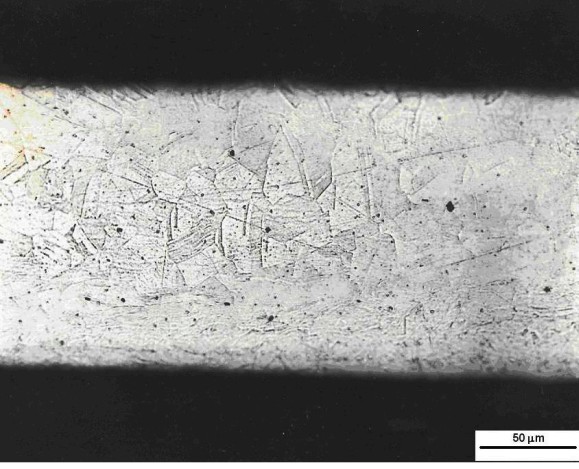

**Figure 6.** The strip blank KT section.

After continuous bending and longitudinal welding using the TIG method, the tubes are a semiproduct for the following drawing operations. The heat impact during the welding process can cause some changes in the heat-affected zone of the weld that influence the plasticity in a negative way. The weld joint is documented in Figure 7a. The same weld joint after the first draw with the floating mandrel and a final tube diameter of 2.84 mm is shown in Figure 7b and after the second draw with a final tube diameter of 2.52 mm in Figure 7c. The geometrical nonuniformity can be seen after seam welding in Figure 7a, but this was eliminated by floating mandrel drawing, as can be seen in Figure 7b,c. No defects were found in the seam welds. Both the microstructure changes of the seam weld and wall thickness reduction were the consequence of the strain generated during the floating mandrel drawing process and they can be seen when comparing Figure 7a–c. The austenitic steels with a carbon content greater than 0.025% and the chromium approx. 18% in the heat-affected zone of the welded joint at low-temperature gradient toward the base material is formed danger of chromium carbide $Cr_{23}C_6$ formation at grain boundaries. These chromium carbide precipitates worsen plasticity during the processing steps with the highest strain occurring during drawing with the floating mandrel. Sufficient protective gas flow during seam welding in protecting camber can be utilized to restrict the creation of chromium carbide precipitates.

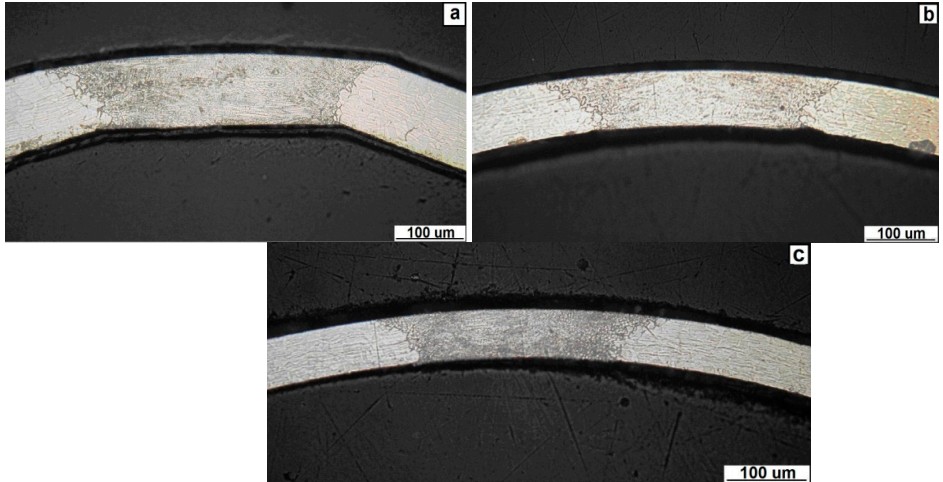

**Figure 7.** The weld joint of the tube from KT after (**a**) welding, (**b**) the first draw and (**c**) the second draw.

The SEM micrographs of the inner side of tubes made of stainless steels produced by EW, and KT are shown in Figure 8. The inner surface of the tube made of stainless steel produced by EW, shown in Figure 8a, is characterized by carbide precipitates in the austenitic matrix creating significant relief. The microstructure of the stainless steel produced by KT, shown in Figure 8b is characterized by a lack of carbide precipitates, but with a coarser grain compared to the steel produced by EW.

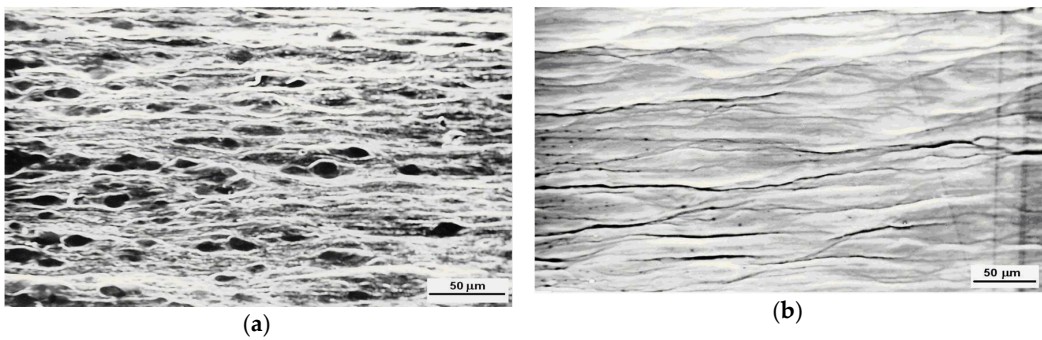

**Figure 8.** The tube inner surface of the material: (**a**) EW 0.65 mm; (**b**) KT 0.8 mm.

Hard carbide precipitates negatively influence formability. They cause misalignment of the outer geometry with the inner circle. The floating mandrel deflects during the drawing process toward the side without carbide precipitates, therefore the wall thickness reduction of the side without carbide precipitates is larger [6]. Significant deformation strengthening in this part leads to plasticity depletion and the creation of conditions for instability during the drawing process with potential breaking of the tube.

### 3.2. The Effect of Tube Eccentricity on the Drawing Process

Geometrical imperfections related to misalignment of the outer and inner diameter of the tube are characterized by eccentricity of it at the center (Figure 9a) [19–22].

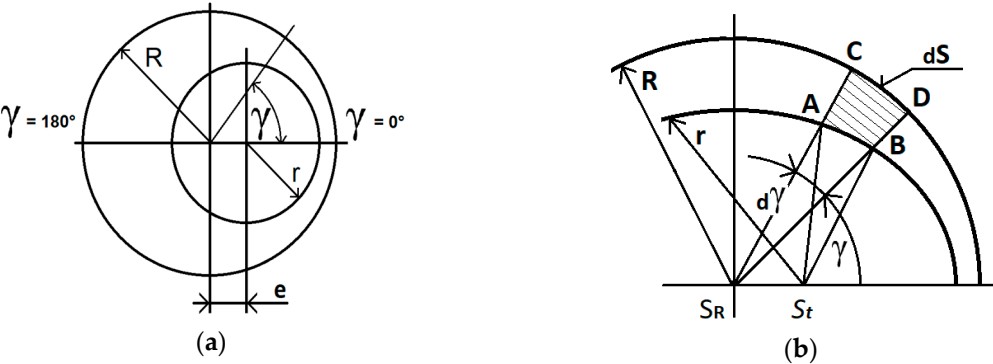

**(a)**　　　　　　　　**(b)**

**Figure 9.** The tube eccentricity (**a**); the elementary surface on cross-section of eccentric tube (**b**).

This imperfection significantly influences both stability during the drawing process and the creation of technological fracture. During drawing of the tubes under the ideal conditions, thus with the eccentricity $e = 0$, the normal stress at the perimeter of tube can be expressed:

$$\sigma_t = \frac{F_t}{S} \text{ resp. } \sigma = \frac{F_t}{\pi \times (R^2 - r^2)} \tag{8}$$

where $F_t$ is the drawing force, $S$ is the cross-section, $R$ is the outer radius and $r$ is the inner radius.

The unit related force on the cross-section of the drawn tube calculated based on the unit angle at the perimeter is:

$$F_1 = \frac{F_t}{2 \times \pi} \text{ or } F_1 = \frac{1}{2} \times \sigma_t \times \left(R^2 - r^2\right) \tag{9}$$

The wall thickness is dependent on angle $\gamma$. Unit force is applied to different planar elements, therefore tensile stress changes continuously depending on angle $\gamma$ (Figure 9b). For this reason, stress $\sigma = f(\gamma)$ is designated $\sigma_\gamma$. Therefore:

$$dF_t = F_1 \times d\gamma \tag{10}$$

The force $dF_t$ affecting an area $dS$ creates stress:

$$\sigma_\gamma = \frac{dF_t}{dS} \tag{11}$$

After appointing of the unit related force:

$$\sigma_\gamma = F_1 \times \frac{d\gamma}{dS} \tag{12}$$

by expression of $F_1$ by tensile stress and cross-section area:

$$\sigma_\gamma = \frac{1}{2} \times \sigma_t \times \frac{\left(R^2 - r^2\right)}{\left(\frac{dS}{d\gamma}\right)} \tag{13}$$

The area $dS$ is according to Figure 9b defined by points $A$, $B$, $C$ and $D$. It can be considered a trapezoid with the height $\Delta r$ which is the immediate wall thickness of the eccentric tube. It is defined as:

$$r = R - \rho \tag{14}$$

where $\rho$ is the inner radius of the planar element; the distance of point $B$ from center $S_R$. The distance of points $\overline{AB}$ (or the length of the arch bordered by points $A$ and $B$) is:

$$\overline{AB} = \rho \times d_y = Z_1 \tag{15}$$

The distance $\overline{CD}$ is defined by radius $R$ and angle $d\gamma$:

$$\overline{CD} = R \times d_y = Z_2 \tag{16}$$

Then, the elementary area of trapezoid $A$, $B$, $C$ and $D$ defined by parameters $Z_1$, $Z_2$ and the height $\Delta r$ is:

$$\Delta S = \frac{1}{2} \times (Z_1 + Z_2) \times \Delta r \tag{17}$$

After substituting of partial results and after modification, the differential area can be expressed as:

$$dS = \frac{1}{2} \times \left(R^2 - \rho^2\right) \times d\gamma \tag{18}$$

The inner radius $\rho$ can be expressed from the triangle defined by points $S_R$, $S_r$, $B$ according to Figure 9b:

$$c_1 = \cos \gamma \tag{19}$$

$$v = e \times \sin \gamma \tag{20}$$

$$c_2 = \sqrt{r^2 - v^2} \tag{21}$$

$$\rho = c_1 + c_2 \tag{22}$$

Through expression of $\rho$ as a function of $\gamma$ we get:

$$\rho = e \times \cos \gamma + \sqrt{r^2 - e^2 \times \sin^2 \gamma} \tag{23}$$

Differentiation of area $S$ with respect to $\gamma$ takes the form:

$$\frac{dS}{d\gamma} = \frac{1}{2} \times \left[R^2 - r^2 - e^2 + 2 \times e^2 \times \sin^2 \gamma - 2 \times e \times \cos \gamma \times \sqrt{r^2 - e^2 \times \sin^2 \gamma}\right] \tag{24}$$

The final value of the normal stress distribution created by the drawing force across a cross-section of the tube with an eccentrical hole in dependent on geometry is:

$$\sigma_\gamma = \sigma_t \times \frac{R^2 - r^2}{R^2 - r^2 - e^2 + 2 \times e^2 \times \sin^2 \gamma - 2 \times e \times \cos \gamma \times \sqrt{r^2 - e^2 \times \sin^2 \gamma}} \tag{25}$$

If we substitute $e = 0$, we get $\sigma_\gamma = \sigma_t$, as the tensile stress for an ideal tube is independent from angle $\gamma$ and is equal to medial normal tensile stress. The behavior of tensile stress across a cross-section is less important than its minimal and maximal value. They will be positioned against each other at angle $\gamma = 180°$. We assumed the maximal value for $\gamma = 0$, then:

$$\sigma_{\max} = \sigma_t \times \frac{R^2 - r^2}{R^2 - (r+e)^2} \tag{26}$$

The minimal stress $\sigma_{\min}$ will be at angle $\gamma = 180°$:

$$\sigma_{\min} = \sigma_t \times \frac{R^2 - r^2}{R^2 - (r-e)^2} \tag{27}$$

If the tensile strength is a boundary value, then the critical value of eccentricity is:

$$e_{\text{crit}} = \sqrt{R^2 - \frac{\sigma_t}{R_m} \times (R^2 - r^2) - r} \tag{28}$$

After depletion of the steel plasticity a fracture can propagate [7].

The position of the floating mandrel during the drawing process changes in the direction of the smallest resistance of the material. This effect is most significant at the first draw, when the material I not influenced by the plastic strain caused by the drawing process. The tube properties will be influenced by these processes:

- Processing of the ribbon;
- Continuous bending of the seam tube;
- TIG welding with a risk of hard carbide precipitation.

These operations can result in the creation of such non-homogenous properties across the cross-section of the tube wall that will cause an eccentrical shape of the cross-section with consequent complexities.

### 3.3. The Technological Process Modeling of Tube Drawing with Limited Dimensions

The knowledge of conventional drawing processes like drawing with a reduction of the wall thickness of the axially symmetric cups was used for the modeling. The drawing of the tubes over a floating mandrel is analogous to deep drawing with a reduction of the wall thickness. The dimensions of the functional parts were designed according to the dimensions of tools accessories when implementing concrete technological processes. The drawing force generation was realized in such a way as to create a face at the beginning of the tube, at the point where there was a drawing machine collet system. The punch affects on this face and generates drawing force acting [23–27]. The models of the individual parts that perform a drawing process with a floating mandrel and the model arrangements are shown in Figure 10a. Drawing without a mandrel is identical to deep drawing of axially symmetric cups in the second draw and the following draws. As in previous cases, it creates a face on the beginning of the tube behind the drawing; die and the model punch generates a drawing force affecting this face. The models of individual parts that perform a drawing process and all model arrangements are shown in Figure 10b. The simulation model was designed with the CATIA V5 R14 software.

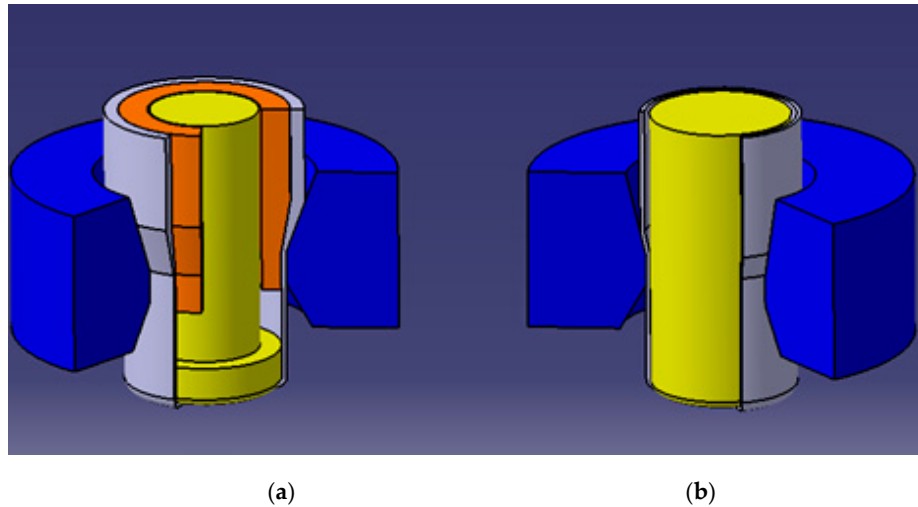

(**a**)　　　　　　　　　　　(**b**)

**Figure 10.** The drawing process modeling. Drawing over floating mandrel (**a**); drawing without a mandrel (**b**).

*3.4. Results of Drawing Simulation of Tubes Drawing with Limited Dimensions*

The results of the simulation were borrowed from the graphics output of the DYNAFORM 5.2 software and are divided into three groups:

STRAIN—drawing force determined through strain;
STRESS—stress in a formed material;
THICKNESS—wall thickness of the tube.

3.4.1. The Values of the Simulation of Drawing Forces

Input data were directly entered from the measured and calculated values analysis of the KT 4306 material. The deciding input data were:

$\sigma_p$—true stress of the formed material at an appropriate true strain;
$R_e$—yield strength of the formed material;
A—percentage elongation of the formed material;
r—normal anisotropy of the formed material;
$\varphi$—true strain during particular technological operations;
$\rho$—density of the formed material;
$\mu$—coefficient of friction;
v—drawing speed of the technological process,
$s_{n-1}$—wall thickness in front of the drawing die;
$s_n$—wall thickness behind the drawing die;

Functional parts geometry—reduction angle of the drawing die, resp. drawing die and mandrel, input and output diameter of the drawing die, diameter of the size fixing area in the drawing die, position of the mandrel against the drawing die.

3.4.2. Boundary Conditions in the First Draw with Floating Mandrel

The boundary conditions for simulations of drawing with a floating mandrel are shown in Table 2.

**Table 2.** The boundary conditions for the simulation of drawing with a floating mandrel.

| $\sigma_p$ (MPa) | $R_e$ (MPa) | $\mu$ (-) | $v$ (m.min$^{-1}$) | $s_{n-1}$ (mm) | $\varphi$ (-) | $d_{n-1}$ (mm) | $d_n$ (mm) |
|---|---|---|---|---|---|---|---|
| 991 | 286 | 0.08 | 10 | 0.1 | 0.4387 | 3.25 | 2.88 |

Figures 11–13 present the results of the drawing force and stress simulation on the material of the tube and wall thickness after tube drawing with a floating mandrel at the first draw from the blank with a diameter of 3.25 mm to a diameter of 2.88 mm and wall thinning from 0.1 mm to 0.075 mm. The values of particular parameters correlating with real values after the technological process are listed in the figures' legends.

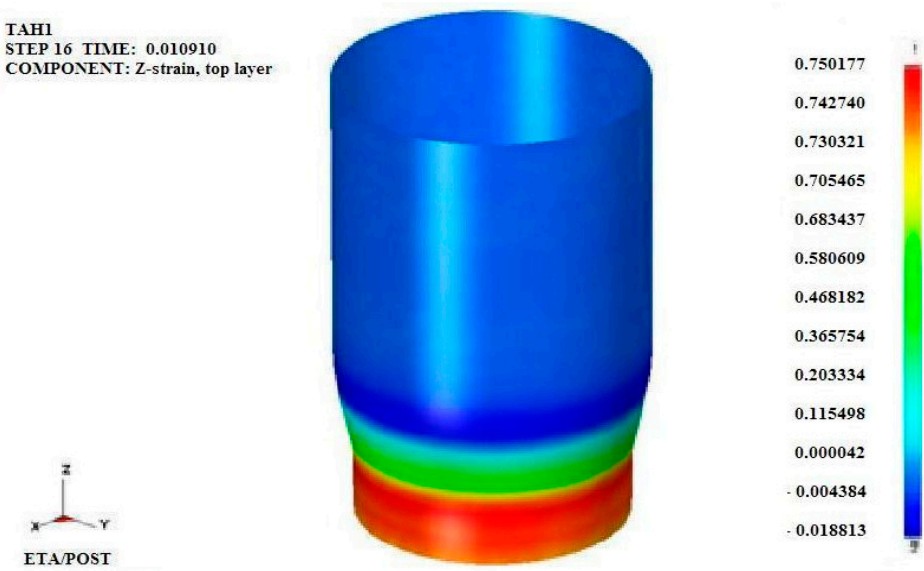

**Figure 11.** Drawing force at the first draw F = 0.740 kN.

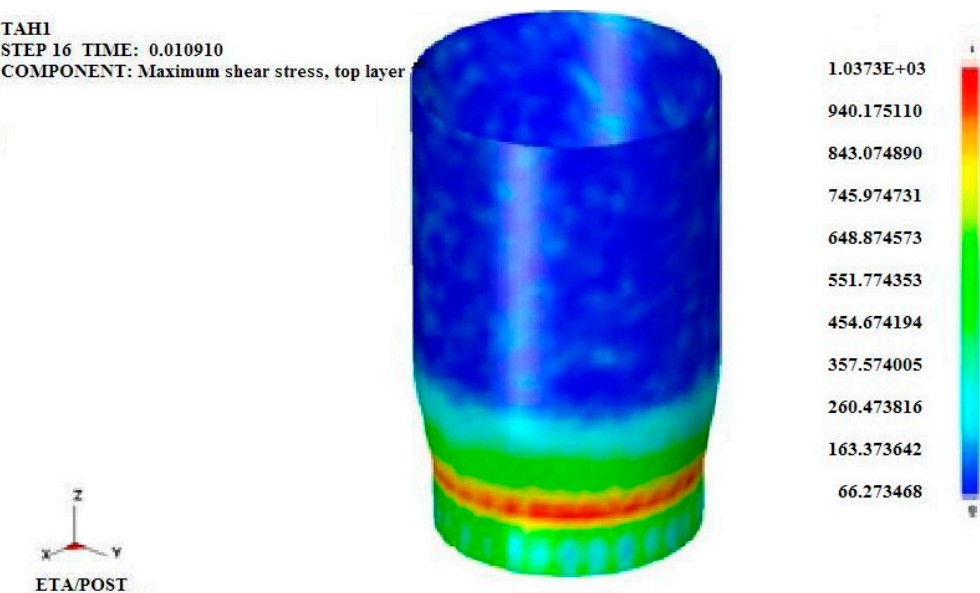

**Figure 12.** Stress in the formed material after the first draw σ = 1 037 MPa.

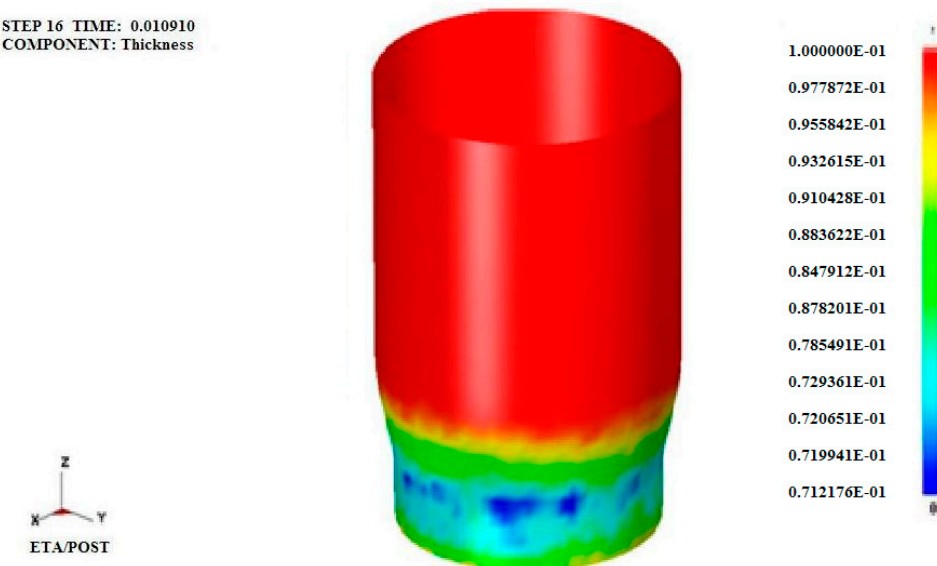

**Figure 13.** Wall thickness after the first draw $s_n$ = 0.0752 mm.

### 3.4.3. Boundary Conditions in the Third Draw without a Mandrel

The boundary conditions for simulations of drawing without a mandrel are shown in Table 3.

**Table 3.** The boundary conditions for the simulation of drawing without a mandrel in the third draw.

| $\sigma_p$ (MPa) | $R_e$ (MPa) | $\mu$ (-) | v (m.min$^{-1}$) | $s_{n-1}$ (mm) | $\varphi$ (-) | $d_{n-1}$ (mm) | $d_n$ (mm) |
|---|---|---|---|---|---|---|---|
| 1215 | 286 | 0.08 | 30 | 0.0634 | 0.1122 | 2.54 | 2.19 |

The second group of simulations of the drawing force, stress and wall thickness after tube drawing without a mandrel after the third draw from a diameter of 2.54 mm to a diameter of 2.19 mm are presented in Figures 14–16. The values of particular parameters correlating with real values after the technological process are listed in the figures' legends.

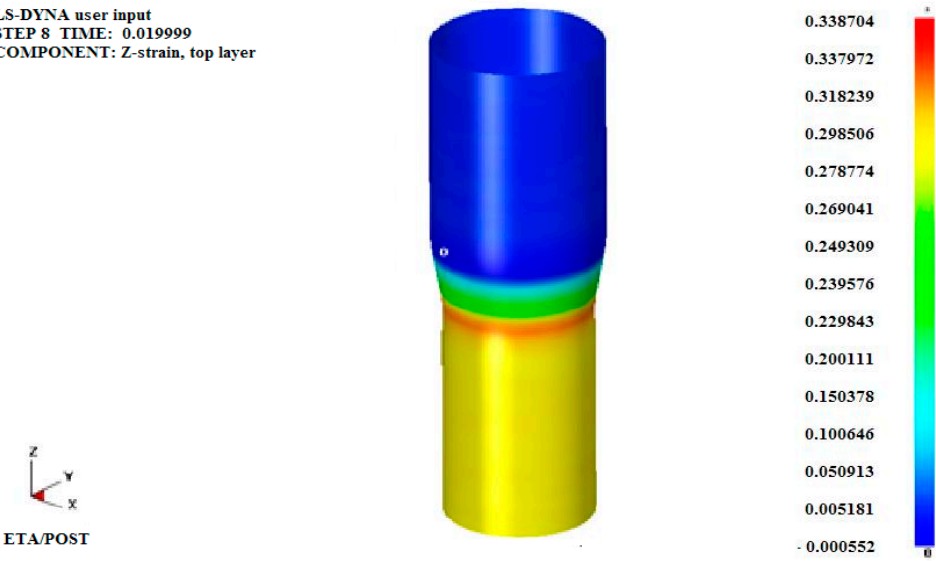

**Figure 14.** Drawing force at the third draw F = 0.310 kN.

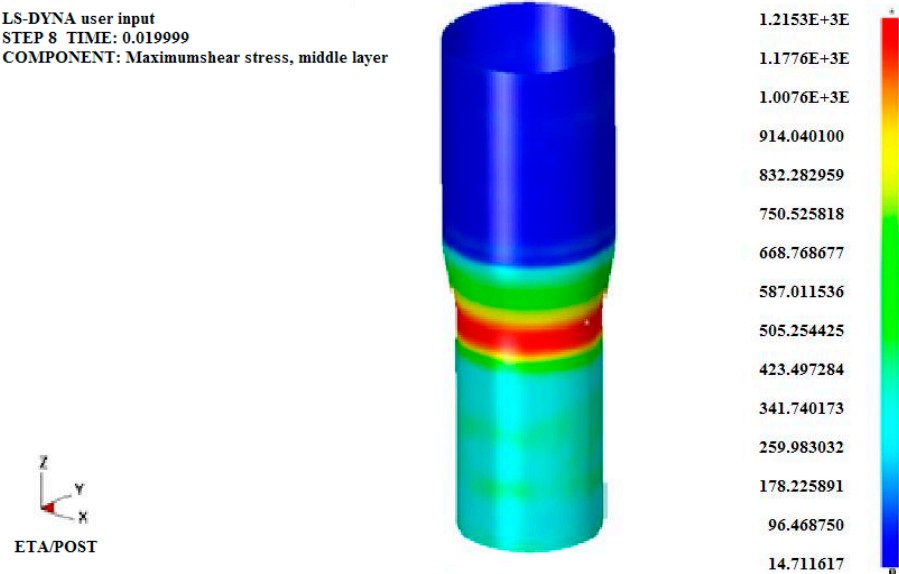

**Figure 15.** Stress in the formed material after the third draw σ = 1 215 MPa.

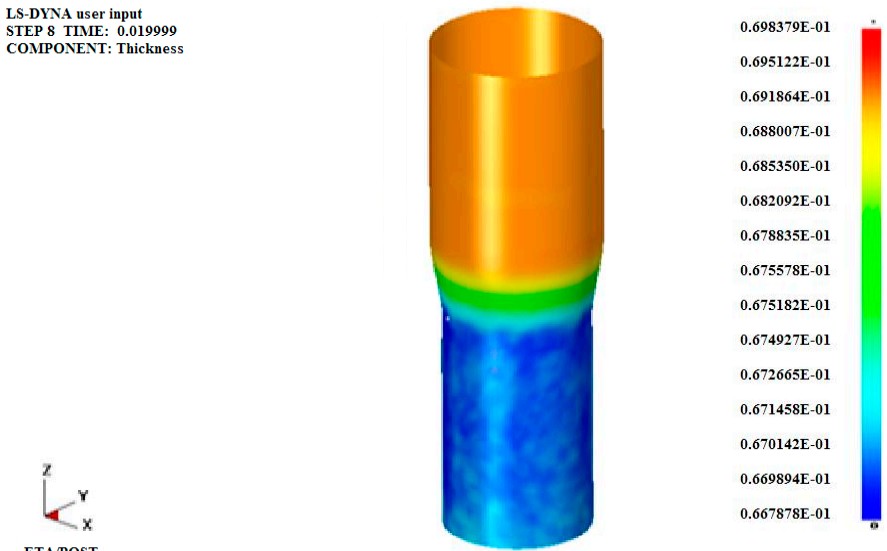

**Figure 16.** Wall thickness after the third draw $s_n$ = 0.0667 mm.

### 3.4.4. Boundary Conditions in the Eleventh Draw without a Mandrel

The boundary conditions for simulations of drawing without a mandrel are shown in Table 4.

**Table 4.** The boundary conditions for the simulation of drawing without a mandrel in the eleventh draw.

| $\sigma_p$ (MPa) | $R_e$ (MPa) | $\mu$ (-) | v (m.min$^{-1}$) | $s_{n-1}$ (mm) | $\varphi$ (-) | $d_{n-1}$ (mm) | $d_n$ (mm) |
|---|---|---|---|---|---|---|---|
| 1947 | 286 | 0.08 | 65 | 0.0673 | 0.4604 | 0.408 | 0.342 |

The third group of simulations of the drawing force, stress and wall thickness after tube drawing without a mandrel after the eleventh draw from a diameter of 0.408 mm to a diameter of 0.342 mm are presented in Figures 17–19. The values of particular parameters correlating with real values after the technological process are listed in the figures' legends.

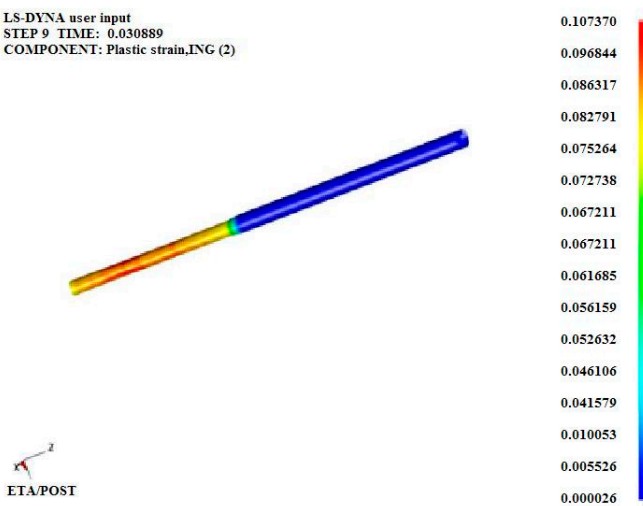

**Figure 17.** Drawing force at the eleventh draw F = 0.090 kN

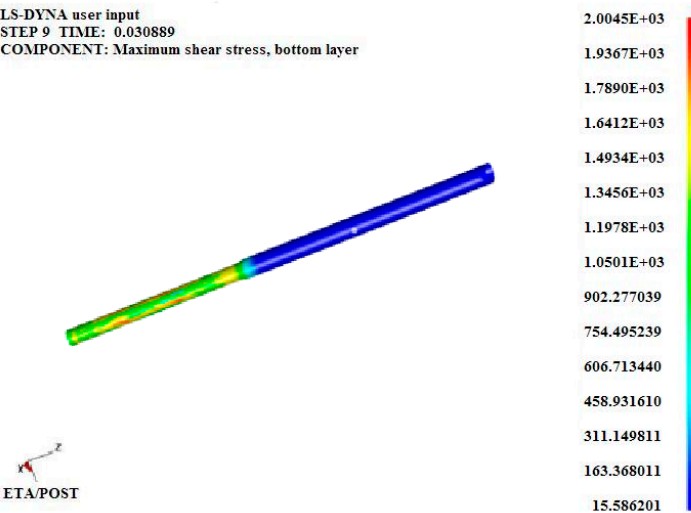

**Figure 18.** Stress in the formed material after the eleventh draw σ = 1 936 MPa.

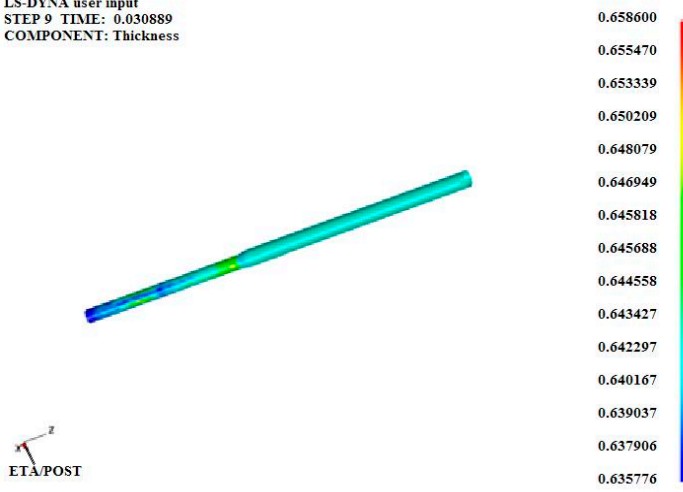

**Figure 19.** Wall thickness after the eleventh draw $s_n$ = 0.0635 mm.

In Table 5, the magnitude of the drawing forces is determined by calculation with the slab method and the stress–strain characteristics of the formed material. These values were compared with the real magnitude of the drawing forces in the technological process which were read from the draw benches Bougrad Toolmatic trolley display device. The calculated values were still compared with forces determined by simulation of individual operations of the whole technological process of tube drawing using the knowledge of classical drawing of axisymmetric drawn cups with or without wall thinning. Table 6 shows the results of the calculated and simulated magnitudes of stresses in the formed material and Table 7 shows the measured and simulated magnitudes of wall thicknesses after each technological operation determined on the basis of strains sizes.

**Table 5.** Comparison table of the calculated, measured and simulated values of drawing forces.

| Draw | Calculated Values | | | Measured Values | | Simulated Values |
|---|---|---|---|---|---|---|
| | $F_{tmin}$ (N) | $F_{tmean}$ (N) | $F_{tmax}$ (N) | $F_{tmin}$ (N) | $F_{tmax}$ (N) | $F_t$ (N) |
| 1st | 588 | 629 | 703 | 520 | 600 | 740 |
| 2nd | 541 | 584 | 670 | 435 | 514 | 680 |
| 3rd | 203 | 215 | 261 | 264 | 287 | 310 |
| 4th | 194 | 207 | 251 | 231 | 268 | 280 |
| 5th | 182 | 191 | 229 | 215 | 249 | 235 |
| 6th | 176 | 187 | 209 | 193 | 229 | 210 |
| 7th | 164 | 171 | 190 | 174 | 211 | 190 |
| 8th | 139 | 148 | 165 | 149 | 167 | 165 |
| 9th | 111 | 117 | 130 | 130 | 155 | 130 |
| 10th | 87 | 99 | 114 | 109 | 115 | 120 |
| 11th | 77 | 85 | 108 | 85 | 93 | 90 |

**Table 6.** Comparison table of calculated and simulated values of effective stresses in the formed material.

| Draw | Calculated Values | | | Simulated Values |
|---|---|---|---|---|
| | $\sigma_{tmin}$ (MPa) | $\sigma_{tmean}$ (MPa) | $\sigma_{tmax}$ (MPa) | $\sigma$ (MPa) |
| 1st | 963 | 991 | 1325 | 1037 |
| 2nd | 1201 | 1209 | 1370 | 1216 |
| 3rd | 1183 | 1215 | 1577 | 1215 |
| 4th | 1333 | 1370 | 1748 | 1345 |
| 5th | 1407 | 1475 | 1863 | 1487 |
| 6th | 1579 | 1598 | 1993 | 1626 |
| 7th | 1621 | 1647 | 2046 | 1487 |
| 8th | 1684 | 1726 | 2129 | 1722 |
| 9th | 1741 | 1757 | 2161 | 1830 |
| 10th | 1833 | 1875 | 2286 | 1843 |
| 11th | 1932 | 1947 | 2360 | 1936 |

**Table 7.** Comparison table of measured and simulated values of tube wall thickness after individual drawings.

| Draw | Measured Values | | Simulated Values |
|---|---|---|---|
| | $s_{ntmin}$ (mm) | $s_{ntmax}$ (mm) | $s_n$ (mm) |
| 1st | 0.0717 | 0.0732 | 0.0752 |
| 2nd | 0.0581 | 0.0634 | 0.0685 |
| 3rd | 0.0661 | 0.0702 | 0.0668 |
| 4th | 0.0593 | 0.0637 | 0.0653 |
| 5th | 0.0605 | 0.0714 | 0.0650 |
| 6th | 0.0575 | 0.0602 | 0.0648 |
| 7th | 0.0643 | 0.0693 | 0.0647 |
| 8th | 0.0696 | 0.0811 | 0.0642 |
| 9th | 0.0720 | 0.0785 | 0.0640 |
| 10th | 0.0596 | 0.0723 | 0.0630 |
| 11th | 0.0540 | 0.0575 | 0.0635 |

## 4. Discussion

The problem of technological fracture formation during the drawing of small diameter tubes was solved by microscopic analysis. During the analysis, attention was paid to the structure of the tested EW and KT materials and their suitability for large deformation associated with the achievement of minimum tube dimensions. The grain size and the presence of phases in the base material as well as in the weld and heat-affected zone, which impair the plasticity of the seam tube material, were monitored. The basic structure of both steels showed an austenitic matrix with a large deformation and marked texture. KT steel had a slightly larger grain compared to EW. In both cases, however, the grain size corresponded to the standard size, which is specified as being up to 0.025 μm. The weld joints showed no defects either in the weld itself or in the HAZ. The microscopic analysis of the longitudinal section of the heat-affected area of tubes with a diameter of 0.65 mm revealed the presence of hard carbide phases along the austenitic grain boundary in the EW material. Their origin can be explained by the pseudobinary diagram $FeCr_{18}Ni_8$-C and the structure of the weld joint and HAZ [6]. Carbide formation is influenced by the C content, which is higher in EW (0.027%) than in KT (0.012%). The width of the carbide formation area $Cr_{23}C_6$ is influenced by the temperature gradient. It can be assumed that their formation and amount will be significantly affected by the cooling intensity in the TIG welding chamber, which can be affected by the circulation of the inert gas Ar. An analysis of stresses and strains during tube drawing over a floating mandrel showed that hard carbide phases can cause the mandrel offset to the side of austenite. Due to the greater thinning of the tube wall, there is a more marked deformation strengthening and an increase in deformation resistance at this point, thereby reducing the plasticity of the steel. Simultaneously, an eccentricity of the outer and inner diameter of the tube is created.

The critical value of eccentricity was defined by mathematical analysis by comparing the stresses in the material on the side of the smallest and largest tube thickness. If the stress on the side of the smallest thickness exceeds the value of the tensile strength of the material, a fracture will occur in the first draw. If this does not occur, the subsequent drawing process for small diameters and wall thicknesses will be undesirably interrupted. This interruption is due to the presence of carbides in the structure and reduced plasticity of the material.

In the analysis of small tube drawing processes, which are semifinished products for injection needles, we also utilized simulations. The results showed:

The magnitude of the drawing forces, which were compared to the calculated and measured values directly from the draw bench;

The stresses in the material determined by simulation and calculation based on true strains and experimentally determined hardening curves for the used material;

Wall thickness sizes for individual draws, which were determined by simulation and by measuring directly from the manufactured tubes.

One of the laws of metal forming, the law of similarity, was used in the simulations. A similar situation of deep drawing of the rotary drawn part with the thinning of the wall in the second draw with tube drawing with a floating mandrel was used. In the same way, the classic deep drawing of the rotary drawn part in the second draw, with tube drawing without a floating mandrel, was used. The simulation models were designed with the CATIA V5 R14 software and the deep drawing processes simulations were run in the DYNAFORM 5.2 software. The boundary conditions were the stress–strain characteristics of KT steel, which were experimentally determined by mechanical tests. For the drawing speeds, the values of the speeds used in real technological drawing processes were entered. The specified value of the friction coefficient was based on a combination of materials (sintered carbide, die; austenite, tube; alloyed hard chrome steel, mandrel) and the lubricant of mineral oil. The inhomogeneity of the material and the eccentricity of the outer and inner tube diameters were not considered in the simulations. The simulation results are presented for the first draw with a floating mandrel and for the third and last draw without a floating mandrel. The tables show the results for the entire process of manufacturing a tube with a diameter of ∅ 0.34 mm and a wall thickness

of 0.057 mm. Overall, the results showed a very good match between the simulations, calculations and the respectively measured values.

## 5. Conclusions

In this presented study of the manufacture of austenitic steel seam tubes which are semifinished products for injection needles, the following were analyzed:

Properties of two materials EW and KT and their suitability in production;

Stress–strain states in the deformation zone and conditions of critical eccentricity of the outer and inner diameter of the tube leading to fracture formation.

Simulation of the drawing process in Dynaform software using the law of similarity between deep drawing of the rotary drawn part with the thinning of the wall in the second draw with tube drawing with a floating mandrel, and also the classic deep drawing of the rotary drawn part in the second draw with tube drawing without a floating mandrel. The results are a comparison of parameters between the values determined by the simulation, and the calculated and measured values of stresses in the material, drawing forces and wall thickness.

Microscopic analysis confirmed that the austenitic matrix is the basic component of the structure of the tested EW and KT steels. In the case of EW steel, consistent with its carbon respect content of 0.027%, the presence of the carbide phase $Cr_{23}C_6$ was confirmed. Due to the large degree of deformation and the small wall thickness, the presence of this phase will significantly affect the plasticity of the material. The assumptions related to stresses in the material affecting the occurrence of the critical eccentricity of the outer and inner tube shape and fracture formation during tube drawing were described and justified mathematically. Analysis of the material structure at the weld site and in the heat-affected zone confirmed that the cooling intensity and the temperature gradient from the weld side to the base material have a significant effect on the deformation properties of the material and the presence of the undesired carbide phase. The formation of the carbide phase is influenced by the temperature gradient in the HAZ. During the very first draw with a floating mandrel, it significantly contributes to the eccentricity formation of the outer and inner shape. Argon, in addition to the function of generating a protective atmosphere, is also used to create a significant temperature gradient. This reduces the formation of areas with an increased chromium concentration thereby reducing the risk of $Cr_{23}C_6$ carbide formation at the grain boundaries. The lower carbon content of X2CrNi19-11 (1.4306) steel with 0.012% C (material KT) results in less risk of $Cr_{23}C_6$ carbide formation. This material is, therefore, more suitable for producing tubes of extremely small diameter and wall thickness, even though this steel has a coarser grain structure. However, it is less sensitive to the formation of carbide phases, which can cause structural inhomogeneity, deterioration of plasticity and eccentricity of the outer and inner tube shape.

The design of the simulation of the tubes drawing, with and without a floating mandrel, was also part of the study. Dynaform simulation software and modeling in Catia software were used. The law of similarities between deep drawing and tube drawing with and without a floating mandrel, together with the entered real boundary conditions of the drawing process simulation, led to results that were comparable with the calculated and measured values of drawing forces, stresses and wall thicknesses. The aim was to show the possibilities of using knowledge from conventional deep drawing processes for nonstandard processes. The achieved results showed very good agreement between simulations, calculations and, respectively, measured values.

**Author Contributions:** A.S. conceived and designed the experiments, and performed modeling, simulations, mathematical analysis and writing. A.B. performed the experiments for the tensile tests and translation. P.Š. performed metallographic samples preparation and translation. A.S., A.B. and P.Š. analyzed the data. Z.G. performed figures design. J.M. performed review and editing. All authors have read and agreed to the published version of the manuscript.

**Funding:** This research received no external funding.

**Acknowledgments:** The authors are grateful for the support of experimental works to the grant agency for the support of the project VEGA 1/0405/19 "Forming and REW joining of hybrid deep drawing parts from high-strength microalloyed blanks and Al-alloy blanks". This work was supported by the University Science Park STU Bratislava ITMS code 26240220084.

**Conflicts of Interest:** The authors declare no conflicts of interest.

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
