# Peer review of "Analysis of the Drawing Process of Small-Sized Seam Tubes"

_metals, doi:10.3390/met10060709_

Round 1

Reviewer 1 Report

The manuscript presents results of theoretical and experimental investigations into drawing of longitudinally welded micro-tubes. The subject is of interest and of practical value. However, major amendments are required before publication:

(1) The overall text, including the title, has to be revised in terms of the used vocabulary, grammar and syntax. In the presented form many statements of the authors are only understandable to a limited extent.

(2) The structure of the manuscript requires improvements: What is the reason for the conducted studies? What is the current state of the art? What are new scientific findings resulting from the presented work (there are new findings and these should be mentioned clearly in the abstract and the summary)? How are the experiments, simulations and analytical equations connected? What is to be shown with the theoretical work?

(3) No reference is made in the text to the literature listed under 9-16 in the section "References".

(4) Instead of mentioning the steel producers of the used materials several times (KRUPP THYSSEN, ERGSTE WESTIG), either the standardized steel designations or designations such as type A, type B should be used. In the presented form it can easily be understood as advertising.

(5) A uniform coordinate system should be used for all representations of technical quantities and results as well as mathematical derivations. Cylindrical coordinates are appropriate here.

(6) The variables used in pictures and equations must be clearly identified in terms of meaning and direction.

(7) The authors describe two drawing processes, one with floating mandrel and one without mandrel. However, in Fig. 10 b it appears that the process they called "without mandrel" was conducted with a mandrel.

(8) The simulation model should be described in more detail. What kind of elements? How many? Why was no material hardening considered and what is the effect on the results? How was the friction value determined? The authors mention that the simulation results corresponds to experimental values. This should be supported by numerical values.

(9) It should be mentioned whether filler material was used for the welding process.

(10) The derivations in equations 1 to 8 can be shortened as it is generally known.

(11) "diameter" in line 248 should be "radius"

(12) Units used in images should be specified and presented components in figures should be designated (e.g. Fig 10).

(13) Important geometric dimensions of the drawing tools should be mentioned (e.g. die angle, etc.).

Author Response

Our replies to the reviewer´s comments and revised version of the article you find in the attachments.

Remarks:

1) The overall text, including the title, has to be revised in terms of the used vocabulary, grammar and syntax. In the presented form many statements of the authors are only understandable to a limited extent.

Our reply: The title is chosen universally because the paper deal with three different problems of small-sized tubes drawing.

2) The structure of the manuscript requires improvements: What is the reason for the conducted studies? What is the current state of the art? What are new scientific findings resulting from the presented work (there are new findings and these should be mentioned clearly in the abstract and the summary)? How are the experiments, simulations and analytical equations connected? What is to be shown with the theoretical work?

Our reply: The reason for solving the task was the production problems of the client. All presented solutions (experiments, simulations and mathematical analysis) led to the removing of problems with the production of small-sized tubes. The client uses the research results in the production process.

3) No reference is made in the text to the literature listed under 9-16 in the section "References".

Our reply: References have been added.

4) Instead of mentioning the steel producers of the used materials several times (KRUPP THYSSEN, ERGSTE WESTIG), either the standardized steel designations or designations such as type A, type B should be used. In the presented form it can easily be understood as advertising.

Our reply: It was necessary for the client of project to determine which of these two materials is more suitable for the production of needles of very small diameters. For this reason, the producer´s designation is used in the text.

5) A uniform coordinate system should be used for all representations of technical quantities and results as well as mathematical derivations. Cylindrical coordinates are appropriate here.

Our reply: We don´t understand the question.

6) The variables used in pictures and equations must be clearly identified in terms of meaning and direction.

Our reply: In our opinion, all parameters are clearly and comprehensibly identifiable.

7) The authors describe two drawing processes, one with floating mandrel and one without mandrel. However, in Fig. 10 b it appears that the process they called "without mandrel" was conducted with a mandrel.

Our reply: Fig. 10 (a,b) illustrate the simulations models of tube drawing over floating mandrel and drawing without mandrel and analogy of ironing and simple deep drawing. The component of yellow colour in both models isn´t a mandrel but a component that generates drawing force. 

8) The simulation model should be described in more detail. What kind of elements? How many? Why was no material hardening considered and what is the effect on the results? How was the friction value determined? The authors mention that the simulation results corresponds to experimental values. This should be supported by numerical values.

Our reply: The results of the measured and calculated values have been supplemented. Because of the scope of article, the simulation conditions weren´t specified. In the case of interest, we can provide the reviewer this information.

9) It should be mentioned whether filler material was used for the welding process.

Our reply: The used welding technology was TIG method where isn´t used filler material.

10) The derivations in equations 1 to 7 can be shortened as it is generally known.

Our reply: Using these equations 1 to 7, the stress-strain characteristics described in Figures 3 and 4 were generated.

11) "diameter" in line 248 should be "radius"

Our reply: The error has been corrected.

12) Important geometric dimensions of the drawing tools should be mentioned (e.g. die angle, etc.).

Our reply: Due to the form of presentation of the results, these parameters weren´t mentioned in the article. Drawing documentation is available.

Reviewer 2 Report

Lines 26-23: Please add some references (where is the process described and/or documented). Also, a flowchart of the process will help the reader to comprehend.

Figure 1 has a low resolution and it is unclear.

Same problem with figure 2 (low resolution).

Please consider replacing the point in the low position “.” with the point in middle position “×” as multiplying operator. Or, at least use the “×” symbol.

Figure 8 b – low resolution and errors in the X axis legend “concetracion” instead of “concentration”

Figures 13, 16, 17 – low resolution

Lines 181-182: “The modelling simulation was realised in CATIA V5 R14 software” – it is not clear here: Was CATIA used for simulation (instead of DYNAFORM) or was used only to build the 3D models presented in figure 10, which were further used for FEM analysis and simulation in Dynaform ?

It is very unclear which are the findings of the research.

Some microstructure analyses of two commercially-available materials were performed. Who requested these analyses? Are these the materials used by the manufacturers of medical needles (if not, why are the medical needles mentioned)? Or are they considered to be used in that purpose, replacing the old ones?

Medical needles seem to be the only application of tube forming at so small diameters. Did the manufacturers of medical needles have problems in selecting the materials for manufacturing them?

What purpose had the FEM simulation and analysis? Did them confirm any experimental research?

Finally, one of the analyzed materials is considered more suitable for drawing (tube forming). Are there any evidences that there is a confusion on the market with regards of his aspect (which material is better)?

Author Response

Revised version of the article you find in the attachment.

Our replies to the reviewer´s comments:

Remarks:

1) Lines 26-23: Please add some references (where is the process described and/or documented).

Our reply: The references were added.

2) Figure 1 has a low resolution and it is unclear. Same problem with figure 2 (low resolution). Figure 8 b and Figures 13, 16, 17 – low resolution

Our reply: All figures have been update.

3) Please consider replacing the point in the low position “.” with the point in middle position “×” as multiplying operator. Or, at least use the “×” symbol.

Our reply: In the formulas, the point has been replaced with the „x“ symbol.

4) Figure 8 b - errors in the X axis legend “concetracion” instead of “concentration”

Our reply: The error has been corrected.

5) Lines 181-182: “The modelling simulation was realised in CATIA V5 R14 software” – it is not clear here: Was CATIA used for simulation (instead of DYNAFORM) or was used only to build the 3D models presented in figure 10, which were further used for FEM analysis and simulation in Dynaform ?

Our reply: CATIA has been used only for modeling, not FEM analysis and simulation of drawing processes.

6) It is very unclear which are the findings of the research.

Our reply: The findigs have been described in Conclusions section in more detail.

7) Some microstructure analyses of two commercially-available materials were performed. Who requested these analyses? Are these the materials used by the manufacturers of medical needles (if not, why are the medical needles mentioned)? Or are they considered to be used in that purpose, replacing the old ones?

Our reply: The analysis has been carried out for the purpose of solving the research project. The analyzed materials are used for production of medical needles.

8) Medical needles seem to be the only application of tube forming at so small diameters. Did the manufacturers of medical needles have problems in selecting the materials for manufacturing them?

Our reply: Táto kombinácia technologického procesu a analyzovaných materiálov je používaná  výrobcom injekčných ihiel, pre ktorého bola výskumná úloha riešená.

This combination of technological process and analysed materials is used by the injection needles producer for whom the research project was solved.

9) What purpose had the FEM simulation and analysis? Did them confirm any experimental research?

Our reply: The simulation results have been compared with the measured and calculated parameters with which they corresponded well.

10) One of the analyzed materials is considered more suitable for drawing (tube forming). Are there any evidences that there is a confusion on the market with regards of his aspect (which material is better)?

Our reply: One of the results of the work is that the material 1.4306 X2CrNi19 -11 with 0.012%C is more suitable for the production of very small-sized tubes because there is no risk of Cr23C6 carbides formation in the structure

Reviewer 3 Report

Dear Authors,

In the manuscript you discuss the subject of forming (by the means of drawing) of austenitic tubes with welded seam. This is a very important issue that is of great practical importance. Although the research and results presented are interesting and necessary, the preparation of the article is insufficient for publication. While reading the article, I had a few comments and remarks that, hopefully, will help shape the final version of your article.

General:

I suggest using the coded markings instead of the full names of the producers of the tested materials: e.g. A and B (or EW and KP). This will facilitate the analysis of the text and prevent a commercial impression.

Remove double spaces. In equations, replace the dots with multiplication signs according to the journal’s guidelines.

Term: "weld thermal field" is extremely rarely used in articles.

The title is very general and on its basis it is difficult to guess the content of the article.

Line 14: The designation of 1.4301 steel is only used in the abstract.

Line 15: „weld line crated” –invalid term.

Lines 19 and 20: „weld thermal field isn´t sufficiently cooled” – invalid wording.

The last sentence of the abstract is surprising, does not match and contains the incorrect term "weld creation". There is a lack of quantitative results.

Keyword: „stability drawing process” is unnecessary or malformed: there is not a single article in Mdpi publishing house, and there are two (2) in Google Scholar with this keyword.

The first paragraph of the Introduction chapter lacks references. Please complete the literature sources from the last two years. Introduction is too short. I suggest you extend it twice. Please state the purpose of the work more clearly.

Line 46: Has the effect of only "weld" or the entire "welded joint" been studied?

Line 71: Remove: „The tungsten electrode is used to creating and keeping electric arc and a gap filling”. This is due to the idea of TIG welding and is obvious. It should be added that autogenous TIG (142) was welded, and parameters: welding current, arc voltage and welding speed, followed by shielding gas flowrate. Was the process carried out automatically?

Line 76: Remove: „when a tungsten electrode is on a negative pole of the voltage source”. It is enough to state: "(DC-)".

Fig. 2 caption: change: „gas jet” to „nozzle” and „non-melting” to „nonconsumable”.

Line 83: The paragraph begins ... surprisingly. Who is illustrated?

Figure 3 and Figure 4 are referred to unnecessarily twice in the text.

Formulas 7 and 8 require further comment.

Line 133: "heat affected zone of the weld". It is best to treat "weld" and "HAZ" as part of a welded joint.

Line 134: „joint after welding "> logical error: after all, each"welded joint" is after "welding".

Line 160: check grammar: „leads”?

Figure 8: the quality is poor. Two typos in „concentration”. Please enter some reference.

Author contributions: it doesn't show who wrote the article. After the research, "review and editing" was carried out immediately. Please correct the typos on lines 326 and 328.

References:

9 of 16 references are older than 20 years. Please find newer wherever possible. For this type of article, the number of references should be about 25 items. I recommend supplementing the literature with English-language articles by authors from various countries, preferably from the last 2 years. This will improve the visibility and impact of the article.

Author Response

Revised version of the article you find in the attachment.

Our replies to the reviewer´s comments:

Remarks:

1) I suggest using the coded markings instead of the full names of the producers of the tested materials: e.g. A and B (or EW and KP). This will facilitate the analysis of the text and prevent a commercial impression.

Our reply: It was necessary for the client of project to determine which of these two materials is more suitable for the production of needles of very small diameters. For this reason, the producer´s designation is used in the text.

2) Remove double spaces. In equations, replace the dots with multiplication signs according to the journal’s guidelines.

Our reply: In the formulas, the point has been replaced with the „x“ symbol.

3) The title is very general and on its basis it is difficult to guess the content of the article.

Our reply: The title is chosen universally because the paper deal with three different problems of small-sized tubes drawing.

4) Line 14: The designation of 1.4301 steel is only used in the abstract.

Our reply: The designation 1.4301 has been added in the text.

5) Line 15: „weld line created” –invalid term.

Our reply:  Term „weld line created” has been replaced with the term „weld line made”.

6) Lines 19 and 20: „weld thermal field isn´t sufficiently cooled” – invalid wording.

Our reply: The wording of this sentence has been modified to: „It is desirable to provide a sufficient inert gas flow and cooling effect during forming a weld joint in a protective atmosphere chamber.“

7) The last sentence of the abstract is surprising, does not match and contains the incorrect term "weld creation". There is a lack of quantitative results.

Our reply: The sentence has been modified.

8) Keyword: „stability drawing process” is unnecessary or malformed: there is not a single article in Mdpi publishing house, and there are two (2) in Google Scholar with this keyword.

Our reply: Keyword: „stability drawing process” has been modified.

9) The first paragraph of the Introduction chapter lacks references. Please complete the literature sources from the last two years. Introduction is too short. I suggest you extend it twice. Please state the purpose of the work more clearly.

Our reply: The introduction has been extended and supplemented with new literary sources. The aim of the work has been formulated more clearly.

10) Line 46: Has the effect of only "weld" or the entire "welded joint" been studied?

Our reply: The entire welded joint has been studied.

11) Line 71: Remove: „The tungsten electrode is used to creating and keeping electric arc and a gap filling”. This is due to the idea of TIG welding and is obvious. It should be added that autogenous TIG (142) was welded, and parameters: welding current, arc voltage and welding speed, followed by shielding gas flowrate.

Our reply: The task of analysis wasn´t to solve the welding parameters. One of the tasks was to solve the problems caused by the welded joint.

12) Line 76: Remove: „when a tungsten electrode is on a negative pole of the voltage source”. It is enough to state: "(DC-)".

Our reply: The sentence has been modified.

13) Fig. 2 caption: change: „gas jet” to „nozzle” and „non-melting” to „nonconsumable”.

Our reply: In Fig.2 have been changed the terms „gas jet” to „nozzle” and „non-melting” to „nonconsumable”.

14) Line 83: The paragraph begins ... surprisingly. Who is illustrated?

Our reply: The sentence has been modified to: „The stress states illustrated in the figures characterize:“

15) Figure 3 and Figure 4 are referred to unnecessarily twice in the text.

Our reply: References to Figure 3 and Figure 4 have been modified.

16) Line 133: "heat affected zone of the weld". It is best to treat "weld" and "HAZ" as part of a welded joint.

Our reply: This comment has been accepted.

17) Line 134: „joint after welding "> logical error: after all, each"welded joint" is after "welding".

Our reply: This comment has been accepted.

18) Figure 8: the quality is poor. Two typos in „concentration”. Please enter some reference.

Our reply: The error has been corrected. The figure has been updated and supplemented with a literary source.

19) Please correct the typos on lines 326 and 328.

Our reply: The errors has been corrected.

20) 9 of 16 references are older than 20 years. Please find newer wherever possible. For this type of article, the number of references should be about 25 items. I recommend supplementing the literature with English-language articles by authors from various countries, preferably from the last 2 years. This will improve the visibility and impact of the article.

Our reply: The manuscript has been supplemented with new literary sources.

Round 2

Reviewer 1 Report

Based on the first review, the authors have significantly improved the quality of the manuscript and implemented well-meant recommendations. The Reviewer thanks for the acceptance of these suggestions. However, there are still points that should be amended before publication:

(a) Necessary corrections concerning the references: (i) The literature should be listed in the section "References" in the order in which it is mentioned in the text, (ii) the reference [19] is not mentioned in the text but should appear there, (iii) the references [21] and [22] are assigned to one single source in the "References" list.

(b) Formula symbols used should be designated directly after these are mentioned (e.g. equations (1) to (7): What is l, r, t, l_0, r_0, t_0 and phi_1, phi_2, phi3).

(c) The authors mention on page 8, line 203, that the drawing force calculated in the simulations is determined from the strain values and show the distribution of strains in Figures 11, 14 and 17, but say in the caption that the drawing force is shown. The theoretical approach for this conversion must be explained.

(d) The strategy referred to as "thin-cut method" on page 13, line 264 is commonly called "slab method".

(e) The term "total stress" used in the text and in Table 6 is probably the so-called "effective stress". This should be checked.

(f) Chapter 4 "Discussion" contains a single subchapter 4.1, where a mathematical derivation is shown. This derivation is not a discussion and the chapter should be called accordingly. In particular, this chapter should be moved before Tables 5, 6 and 7, as these tables use the values calculated with this derivation.

(g) It is suggested to use in chapter 4 the term "related force" instead " unit applied force".

(h) The entire text should be proofread by a native English speaker, since despite extensive improvements, several points still need to be revised with regard to grammar and vocabulary.

Author Response

Thank you for your remarks.

Remarks:

1) Necessary corrections concerning the references: (i) The literature should be listed in the section "References" in the order in which it is mentioned in the text, (ii) the reference [19] is not mentioned in the text but should appear there, (iii) the references [21] and [22] are assigned to one single source in the "References" list.

Our reply: (i) The literature has been listed in the order as it is mentioned in the text. (ii) The reference [19] has been inserted in the text (line 199). (iii) The references [21] and [22] have been modified.

2) Formula symbols used should be designated directly after these are mentioned (e.g. equations (1) to (7): What is l, r, t, l_0, r_0, t_0 and phi_1, phi_2, phi3).

Our reply: These symbols have been added and described (page 4).

3) The authors mention on page 8, line 203, that the drawing force calculated in the simulations is determined from the strain values and show the distribution of strains in Figures 11, 14 and 17, but say in the caption that the drawing force is shown. The theoretical approach for this conversion must be explained. 

Our reply: The drawing forces determined by simulation were realised in DYNAFORM 5.2 software. The procedure and method of simulation were based on the values of the magnitude of the stresses in the tube material and instantaneous size of the wall thickness of the tube which were also determined by simulations for individual draws.

4) The strategy referred to as "thin-cut method" on page 13, line 264 is commonly called "slab method".

Our reply: The term "thin-cut method" has been replaced with the term "slab method".

5) The term "total stress" used in the text and in Table 6 is probably the so-called "effective stress". This should be checked.

Our reply: The term "total stress" has been replaced with the term "effective stress".

6) Chapter 4 "Discussion" contains a single subchapter 4.1, where a mathematical derivation is shown. This derivation is not a discussion and the chapter should be called accordingly. In particular, this chapter should be moved before Tables 5, 6 and 7, as these tables use the values calculated with this derivation.

Our reply: The subchapter 4.1 „The effect of tube eccentricity on drawing process” has been moved before Tables as the subchapter 3.2 (page 8).

7) It is suggested to use in chapter 4 the term "related force" instead " unit applied force".

Our reply: The term "unit applied force" has been replaced with the term "related force".

Reviewer 2 Report

Authors provided answers and solutions to all issues risen by the reviewer.

Consequently, I consider that now the work is suitable for publication in the journal.

Author Response

Thank you for your remarks.

Reviewer 3 Report

Dear Authors, 

Thank you for the answers.

I understand that the client of the project may have their expectations regarding the test raport. But a scientific article in a reputable journal addressed to a wider audience cannot be subject to the same rules as commercial work focused on the satisfaction of a selected customer.

"weld line" is a term mainly used for welding of polymers. Maybe it is a "fusion line"? That is, the border between the welded area and HAZ, characterized by partial melting of the grains.

Author Response

Remark:

"weld line" is a term mainly used for welding of polymers. Maybe it is a "fusion line"? That is, the border between the welded area and HAZ, characterized by partial melting of the grains.

Our reply: Thank you for your remarks. As form the term "weld line", this term is also used for joining metallic materials. This issue was consulted with a professor who has been working in the field of welding a long time. Based on his recommendation, we used this term in the article.